# Impact of Potassium Pre-Harvest Applications on Fruit Quality and Condition of Sweet Cherry (*Prunus avium* L.) Cultivated under Plastic Covers in Southern Chile Orchards

**DOI:** 10.3390/plants10122778

**Published:** 2021-12-16

**Authors:** Marco Bustamante, Ariel Muñoz, Iverly Romero, Pamela Osorio, Sergio Mánquez, Rocío Arriola, Marjorie Reyes-Díaz, Alejandra Ribera-Fonseca

**Affiliations:** 1Centro de Fruticultura, Facultad de Ciencias Agropecuarias y Forestales, Campus Andrés Bello, Universidad de La Frontera, Avenida Francisco Salazar 01145, Temuco P.O. Box 24-D, Chile; marco.bustamante@ufrontera.cl (M.B.); ariel.munoz@ufrontera.cl (A.M.); 2Instituto de Investigaciones Agropecuarias (INIA), Carillanca Station, km 10 Camino Cajón-Vilcún, Temuco P.O. Box 929, Chile; iverlyromero@gmail.com; 3Research, Development and Innovation Department, Exportadora Rancagua S.A.—Ranco Cherries, Route 5 South, 04000, km 80, Rancagua P.O. Box 576, Chile; posorio@ranco.cl (P.O.); smanquez@ranco.cl (S.M.); rarriola@ranco.cl (R.A.); 4Departamento de Ciencias Químicas y Recursos Naturales, Facultad de Ingeniería y Ciencias, Campus Andrés Bello, Universidad de La Frontera, Avenida Francisco Salazar 01145, Temuco P.O. Box 24-D, Chile; marjorie.reyes@ufrontera.cl; 5Center of Plant-Soil Interaction and Natural Resources Biotechnology, Scientific and Technological Bioresource Nucleus (BIOREN), Campus Andrés Bello, Universidad de La Frontera, Avenida Francisco Salazar 01145, Temuco P.O. Box 24-D, Chile

**Keywords:** foliar fertilization, cracking, firmness, acidity

## Abstract

In rainy locations, sweet cherry is cultivated under plastic covers, which are useful to prevent fruit cracking but decrease cherry quality such as firmness and acidity. Here we evaluate the impact of pre-harvest K foliar applications on harvest and post-harvest fruit quality and condition of sweet cherry cultivated under plastic covers in southern Chile orchards. The study was performed on two commercial orchards (cv. Regina), located in different regions, during two consecutive seasons. In all cases, a conventional K regime (four sprays) was compared to an intensive K regimen (seven sprays). Results showed that cherries from the most southern region revealed lower acidity but higher soluble solids content weight and size. The intensive K regime improved the firmness and acidity of fruits of covered trees at harvest and post-harvest. Moreover, we found that condition defects were higher in fruits from un-covered trees and that trees grown under intensive K regime showed lower levels of cracking at harvest and pitting at post-harvest compared to trees treated with the conventional K regime. Otherwise, pedicel browning was inconsistently affected by K sprays. Our results revealed that an intensive K regime could improve the quality and condition of fruits at harvest and post-harvest in covered orchards of sweet cherry cv. Regina; however, the impacts can significantly vary depending on season and locality.

## 1. Introduction

Sweet cherry (*Prunus avium* L.) is one of the most important fruit crops cultivated in temperate climates. A strong increase in the production of sweet cherry orchards worldwide has occurred, especially in the last two decades, mainly due to high consumer demand and good grower returns [1]. As a result, this fruit crop has expanded even into regions where it has not been a conventional crop and where adverse weather conditions limit sweet cherry yield and quality.

The production of sweet cherry in Chile increased from 41,000 to 234,000 tons between 2009 and 2019. In this same period, the country’s share of the world market has grown from 1.9% to 9.0%, placing our country in third place among the countries with the highest sweet cherry production, after Turkey and the United States of America, which also reached the first place in the international export ranking [2,3]. According to the Fruit Exporters Association of Chile [3], Chile exported 84% of its cherry volume to the Chinese market. A recent report by the U.S. Department of Agriculture’s Foreign Agricultural Service, the sweet cherry planted area in Chile is estimated to increase by 11.5% in the 2021/22 season, reaching a total of 44,000 hectares (producing around 395,000 metric tons, which are mainly concentrated in General Bernardo O’Higgins (VI) and Maule (VII) Regions [4].

The counter-season cherry production of Chile with respect to China, the main importer country of Chilean sweet cherries, the existence of trade agreements between both countries, and the sustainable economic growth of this Asiatic country [5,6] have favored the increase of the planted area in Chile. However, the explosive increase of sweet cherry worldwide offerings, the consequent stagnation or decrease of market prices for this fruit [7], and the mega-drought that affected the largest sweet cherry production area of South America [8,9] have led producers to look for new cultivation areas in order to obtain better prices for this fruit crop [10]. In this context, the southern zone of Chile (from La Araucanía to Los Lagos) [11] is currently considered an attractive new place for sweet cherry production due to the possibility of producing mid-maturing and late-maturing varieties, which can arrive in China during the Chinese New Year or the Spring Festival, these being among the most anticipated dates for exporters of Chilean cherries. It is well-known that some climates have advanced harvests in the early cultivars or delayed harvests in the late cultivars, with obvious market benefits [12].

Sweet cherry production in the Southern macro-zone of Chile (37°35′–40°33′ S) has several climatic constraints, including late spring frost but mainly high precipitation levels at bloom and pre-harvest. This zone has a rainy temperate climate with annual rainfall ranging from 1200 to 2800 mm per year, which partially decreased through the productive season [13]. One of the main limitations to sweet cherry production is rain-induced fruit cracking [14]. The fruit cracking is induced by the cuticle transpiration of the fruit and/or via evapotranspiration stream from the roots. The susceptibility of fruits to this problem increases during the last period of cherry development, shortly before harvest, and particularly after (or during) rainfall [15]. For this reason, sweet cherry producers of this zone frequently prefer to produce sweet cherry varieties resistant to fruit cracking, such as Regina [7,16,17]. Also, most sweet cherry producers of southern Chile need to use plastic covers to protect sweet cherry orchards from the flower falls and fruit cracking provoked by rains [18,19]. The use of plastic covers has been reported to be an effective way to reduce rain-induced cracking, since the covers present a physical barrier that prevents direct water contact with the fruit surface [20].

Besides representing a physical barrier that prevents direct water contact with the fruit surface, plastic covers can modify the micro-environmental conditions of the orchards, increasing both air temperature (by almost 15 °C) and relative humidity [21,22], thus protecting fruit crops against frost but increasing the incidence of fungal diseases [23]. Previous works have shown that plastic covers can negatively alter fruit quality parameters such as firmness (decreases) and caliber (increases) [21,24,25], decreasing the resistance of fruits to mechanical damage that can occur during harvest, handling, and a long post-harvest trip. In addition, a notorious decrease in the crispiness sensation, an attribute desired by consumers, has been lately reported [25,26]. Moreover, deficient skin color and low acidity levels have been detected in sweet cherries produced in several southern Chile locations, probably explained by reduced radiation under plastic covers and the low thermal sum of this zone [27]. Therefore, it is urgent to find agronomical strategies to mitigate the adverse effects of weather conditions and plastic covers on the quality parameters of cherry fruit cultivated in southern Chile.

Potassium (K) pre-harvest foliar applications have emerged as an alternative to improve firmness, solid soluble contents, and acidity levels in the fruits of several fruit species [28,29,30]. Previous studies have shown that adequate K nutrition increased the quality attributes of fruits, especially size and shape, improving soluble solids contents, ascorbic acid concentrations, fruit color, shelf life, and also shipping quality in several fruit crops, including muskmelon [31,32] cantaloupe [33], and tomato [34]. Indeed, Stiles [35] and Hoying [36] agree that fruits could be smaller, have dull colors, and be tasteless in K-deficient fruit trees due to insufficient acid levels and be thick-skinned. According to Nava [37], K fertilization has positive impacts on fruit color and total soluble solid content but negatively affects the flesh firmness of apples. Likely, it is reported that foliar K sprays improve the firmness, weight, sweetness, and acidity of Red Delicious apples [38], and apricots [39]. Yener and Altuntaş [28] demonstrated that adequate K fertilization increased fruit firmness, caliber, weight, soluble solids content, and acidity in sweet cherries cultivated in Turkey. Additionally, Ucgun [40] concluded that excessive K fertilization did not positively impact cherry fruit quality and might even have negative impacts.

According to our knowledge, there are no reports about the behavior of quality and condition of fruits from trees cultivated under plastic covers in response to K foliar fertilization programs. Therefore, this work was aimed at evaluating the impact of pre-harvest K foliar applications on harvest and post-harvest fruit quality and the condition of sweet cherries cultivated under plastic covers in southern Chile orchards.

## 2. Results

### 2.1. Quality and Conditions of Fruits at Harvest

Our results about the impact of treatments on fruit weight revealed that, in the Perquenco orchard, the weight of cherries from un-covered trees was significantly lower than the weight of cherries harvested from trees protected with cover; nonetheless, this effect was detected only in the 2019/2020 season, without differences observed in the 2018/2019 season. Regarding weight variation between seasons, no significant differences were observed for any treatment at the Perquenco orchard. Furthermore, we did not find differences in fruit weight in both locations between the different K fertilization regimes. In general, fruits produced in the Puerto Octay orchard exhibited higher weight compared to fruits from covered trees from the Perquenco orchard, especially during the 2020/2021 harvest season (Table 1).

Concerning fruit size (caliber measured as equatorial diameter), the results indicated that fruits produced in Perquenco orchards in the covered plot were larger (~1.6 mm) than those from un-covered trees; however, this impact was detected only in the 2019/2020 season. Additionally, our study showed that in the Perquenco orchard, in the 2018/2019 season, the size of fruits from trees treated with intensive K fertilization was larger (~0.5 mm) than fruits from trees grown under the conventional K regime, while non-differences for this parameter were observed in the 2020/2021 season. Also, and in agreement with those observed for fruit weight, the caliber of fruits harvested in Puerto Octay was greater compared to those of fruits from covered trees of Perquenco orchard; however, these differences were only detected during the 2020/2021 season, whereas non-differences among localities were observed in the 2019/2020 season (Table 1).

In the Perquenco orchard, significant differences were found for fruit firmness between covered and un-covered trees; these differences were detectable in both study seasons (2018/2019 and 2019/2020). In this location, under the conventional K foliar regime, the use of plastic cover caused a decrease in fruit firmness compared to those observed in the un-covered plot. These differences reached up to 27% when comparing fruits from un-covered trees subjected to the intensive K regime (K+) and those from covered trees grown under the conventional K regime (K−). On the other hand, the intensive K regime, independent of plastic cover, significantly improved cherry firmness, but mainly in covered trees, mitigating the negative impact of the covers on this quality parameter. In fact, during the 2018/2019 season, the firmness of fruits from un-covered trees grown under the conventional K regime was comparable to those detected in fruits from covered trees subjected to the intensive K regime. During the 2019/2020 season, no significant differences for fruit firmness were found between the cover and K fertilization treatments in the same locality (Table 1).

Regarding the Puerto Octay orchard, fruits from trees grown under the conventional K regime showed higher firmness than those from the Perquenco orchard, but only in the 2019/2020 season, while firmness in the 2020/2021 season of fruits from both localities was very similar. Moreover, as occurred in the Perquenco orchard, fruits harvested from trees treated with the intensive K regime were firmer compared to cherries from trees grown under the conventional K regime; however, this impact was only detected during the 2020/2021 season, whereas, in the 2019/2020 season, the effect was opposite. The statistical analysis of this study revealed significant interactions between seasons and localities for almost all treatments (Table 1). In general, no differences were observed for fruit firmness among locations nor between seasons in the same locations (Table 1). In addition, concerning the effect of the fruit location in the canopy on fruit quality, our results showed that, independent of the cover and of K fertilization treatments, fruits harvested from the lower part of the trees showed higher levels of firmness compared to those collected from the upper part of the trees (Figure 1a).

Concerning cherry sweetness, our observations in the Perquenco orchard showed that, independent of the cover and K treatments, total soluble solids (TSS) were higher in cherries produced in the 2019/2020 season than those harvested in the 2018/2019 season. Also, we found that TSS tended to be lower in fruits produced in covered trees than un-covered ones; nonetheless, significant differences were only detected in the 2019/2020 season. In this case, fruit from the un-covered plot showed at least 1.7 Brix more than those from trees protected with the cover. Furthermore, results revealed that the treatments with the intensive K regime increased the TSS of fruits compared to cherries produced in trees grown under the conventional K regime; however, this impact was only observed during the 2018/2019 season. In addition, data showed that TSS was higher in fruits produced in Puerto Octay than those produced in Perquenco. In contrast to those observed in the Perquenco orchard, independent of the season, the application of additional sprays of K did not affect the TSS of fruits produced in the Puerto Octay orchard (Table 1). Similarly to those detected in Perquenco, TSS of Puerto Octay fruits harvested in the 2019/2020 season were higher compared to fruits collected in 2020/2021.

Otherwise, our results regarding variations in the titratable acidity (TA) of fruits at harvest indicated that, in the Perquenco orchard, fruits from un-covered trees exhibit higher (by around 14%) TA values than fruits harvested from covered trees. These differences were observed in both seasons (2018/2019 and 2019/2020). Additionally, we found that the TA of cherry produced in trees grown under the intensive K regime was higher than those from trees cultivated under the conventional K regime; nevertheless, significant improvements were only observed in the covered plot. On the other hand, in the Puerto Octay orchard, significant differences for fruits TA were only detected between both fertilization regimes during the 2021 season. In this case, the intensive K fertilization regime increased TA (by around 33%) with respect to fruits produced in trees grown under the conventional K fertilization treatment (Table 1). When comparing fruit TA between season-localities interactions, it was possible to identify a significant effect of the locality factor in this parameter. Therefore, in the Perquenco orchard, TA of fruits ranged from 0.84 to 1.08% of malic acid, significantly exceeding those detected in fruits produced in the Puerto Octay orchard that exhibited low TA values ranged from 0.45% to 0.69% of malic acid, considering both study seasons. Regarding the variation between seasons, no differences were found between both harvests in Perquenco; nonetheless, in the Puerto Octay orchard and independent of the treatment applied, fruits produced in the 2019/2020 season exhibited higher TA than cherries produced in the 2020/2021 season (Table 1). Finally, when we compare fruits from both locations in the canopy (upper area versus lower area), independent of the location and the cover and K treatments applied, no significant differences were found for fruit TA (Figure 1b).

Finally, fruit maturity index (MI) results revealed higher values in Puerto Octay fruits than fruits produced in the Perquenco orchard, independent of the cover and K treatments applied. The lower values of TA explain this observation and, to a lesser extent, to the higher values of TSS detected in the assay of Puerto Octay fruits. Also, data showed significant differences between K treatments in fruits produced only in the Perquenco orchard during the 2018/2019 season and in Puerto Octay during the 2020/2021 season. In the Perquenco orchard, fruits from un-covered trees grown under the conventional K regime presented the highest MI values compared to the other treatments. In the same orchard, fruits from un-covered trees grown under the intensive K regime showed higher MI than those cultivated under the conventional K regime. In contrast, the application of additional sprays of K to trees grown in the Puerto Octay orchard cause a significant decrease in MI values compared to those detected in fruits from trees grown under the conventional K regime during the 2020/2021 season, while in the 2018/2019 season, although no significant difference was found, there was a slight decrease in this parameter. Moreover, results showed no significant differences for MI values between fruits harvested from the upper and lower areas of the canopy of trees grown in the Perquenco orchard. However, the MI of fruits collected from the upper area of the canopy of trees of the Puerto Octay orchard was lower than the fruits of the lower area of the canopy, which is probably closely linked to the higher levels of TSS in fruits harvested from the upper area of the canopy; however, these differences were only detected in the 2019/2020 season (Figure 1c).

A principal component analysis (PCA) provided information about the most determinant parameters that varied in response to studied factors: locality, season, use of covers and K fertilization regime, allowing a partial visualization of the data set in two dimensions corresponding to the two main components with the highest incidence (Figure 2). The scatter plot (Figure 3a) shows the six more relevant quality parameters for sweet cherry fruit. Results of PCA revealed that the first principal component (PC1) accounted for ~64.6% of the variance, while principal component 2 (PC2) accounted for ~22.4% of the variance, indicating a high differential response of fruit quality in response to the different treatments (Figure 2A). In addition, the loading plots (Figure 2B) indicated that PC1 is moderately correlated with all fruit parameters evaluated, exhibiting positive correlations with weight (r = 0.49), size (r = 0.43) and maturity index (r = 0.39) but negative correlations with the firmness (r = −0.41), TSS (r = −0.25) and TA (r = −0.44) of cherry fruits.

Moreover, PC2 exhibited high positive correlations with TSS (r = 0.7) and maturity index (r = 0.5) but low positive correlations with fruit size (r = 0.27) and firmness (r = 0.16); there were also moderate negative correlation with TA (r = −0.4) and weight (r = −0.02). On the other hand, PCA allowed the identification of four groups defined by their differential response to the treatments based on the fruit quality characteristics: Group 1, composed of the treatment consisting of un-covered trees grown at Perquenco in both seasons (3, 4, 7, 8), which had a high positive correlation with the firmness and TA of fruits but a high negative correlation with MI, size, and weight; Group 2, composed of treatments on covered trees grown in Puerto Octay during the 2020/2021 season (11, 12), which exhibit a high correlation with the parameters weight and MI but negative correlation with fruit firmness; in general, the variance of this group is highly correlated with PC1. The relative position of treatment in this PCA consists of covered trees subject to the intensive K fertilization regime (12) and respect to the treatment consisting of covered trees grown under the conventional K regime (11), suggesting that the negative effects of the use of cover can be attenuated with the use of intensive K fertilization. Group 3, composed of covered trees grown in Puerto Octay during the 2019/2020 season (9, 10), and revealed a low correlation with PC1, while PC2 has only a slight positive correlation with TSS and a negative correlation with size. Group 4, composed of the treatments of covered trees grown in Perquenco in both seasons (2018/2019 and 2019/2020) (1, 2, 5, 6); the variance of this group is poorly explained by PC1, while, regarding PC2, it exhibited a slight positive correlation with the caliber and a negative correlation with the TSS. In summary, PC1 allowed the discrimination of two groups of highly opposite behavior in terms of fruit quality parameters (Groups 1 and 2), for which the most representative parameters factors were firmness, weight, and TA; whereas PC2 allowed discrimination of Groups 3 and 4, with TSS and size the most important parameters.

In the present study, an analysis of the proportion of variance explained was applied (Table 2) to complement the information provided by the PCA. This analysis allows the quantification of the incidence of the studied factors on the quality parameters of the fruits. According to the results, in Perquenco, the factor “use of covers” revealed the greatest incidence on the variance on most of the quality parameters, explaining 45% of the variance for fruit firmness and at least 24% of the variance for weight, size, and TA of fruits. Moreover, “the season” factor had a strong influence on the variance of TSS (24.5%) and MI (43.5%). On the other hand, although the “K treatment” factor had a significant effect on some quality parameters (Table 2), the percentage of variance explained for this factor was 4.2% for size, the parameter on which it had the greatest impact. Finally, the factor “canopy area” only had a significant effect on TSS (11.2%) and, consequently, but to a lesser extent, on the MI (3.8%). Regarding the proportion of variance explained in the Puerto Octay orchard, the “season” factor had the greatest effect on all quality parameters studied, explaining from 24.7% (for fruit size) to 51.9% (for fruit TA) of the variance. Otherwise, in this locality, the “K treatment” factor strongly affected fruit TA and consequently on the fruit MI, explaining 12.3% and 18.2% of the data variance, respectively. Finally, the factor “canopy area” revealed a significant influence on fruit firmness (13.9%) and TSS (17.2%).

In addition to the fruit quality analyses, the incidence of the most frequent condition defects in sweet cherry fruit at harvest was quantified (Figure 3). The results revealed that the major defect independent of the locality, season, and K treatments was fruit cracking, followed by fruit pitting and pedicel browning. Data from Perquenco plots indicated that condition defects were always higher in fruits produced by un-covered trees than by covered ones. Moreover, we found that increased K application reduced the incidence of fruit cracking, which was observed in both study sites and all seasons and mainly in fruits from the un-covered trees of Perquenco. Indeed, in the 2019/2020 season, in the Perquenco orchards, the cracking incidence of fruits harvested from trees protected with plastic cover was significantly lower than those observed for fruits from un-covered trees. Thus, cracking levels in fruits decreased from 20.5% to 3.25% and from 8.75% to 1.75% when K supply in un-covered trees was increased from four (conventional regime) to seven (intensive regime) applications. In addition, our results indicated that fruits produced in the covered trees of Puerto Octay exhibited higher cracking incidence than the fruits produced on the covered trees of Perquenco.

Also, results indicated that fruits grown under the intensive K regime showed less pedicel browning than cherries produced in trees cultivated under the conventional K regime. Nonetheless, this positive impact was only detected in 2018/2019, whereas no differences were found in 2019/2020. In contrast, in the Puerto Octay orchard, fruits grown under treatment with higher K doses showed greater incidence of fruits with brown pedicels; however, this effect was detected only in 2019/2020. In general, pedicel browning incidence in fruits from covered trees was higher in the Puerto Octay orchard compared to the Perquenco orchard. Finally, regarding fruit pitting, results indicated that the incidence of this defect in fruits produced on covered trees was also higher in the Puerto Octay orchard than the Perquenco site, tending to be slightly higher in fruits from un-covered trees than covered trees in Perquenco. Additionally, we found that fruits from covered trees at both localities treated with the intensive K regime exhibited higher pitting incidence compared to fruits produced in covered trees cultivated under the conventional K regime. Nevertheless, these differences were observed only in 2019/2020 for both localities, whereas in the rest of the seasons and in the uncovered trees of Perquenco, no differences were found.

### 2.2. Quality and Condition of Fruits at Post-Harvest

Results about the behavior of fruit quality parameters during post-harvest storage showed that the firmness of cherries produced in the Perquenco orchard at post-harvest was higher in fruits from un-covered trees and that the increase of K applications from four to seven sprays increased the fruit firmness of fruits from covered and un-covered tress; however, this impact was detected only in the 2018/2019 season (Table 3).

In the same location, results showed that TSS at post-harvest were equal or higher in fruits produced by un-covered trees compared to covered trees and that fruits from trees grown under the intensive K regime exhibited higher TSS than fruits from trees grown under the conventional K regime. Nonetheless, these differences were detected in both locations in the 2018/2019 season, whereas in the 2019/2020 season, non-variations were found (Table 3).

Furthermore, our results indicated that the TA of fruits was strongly reduced during post-harvest compared to the values detected at harvest. Also, here we found that, in the Perquenco orchard, fruits grown under plastic covers exhibited equal or inferior AT levels compared to fruits from un-covered trees. In addition, data of Perquenco plots revealed that TA values were significantly higher in the 2019/2020 season compared to those reached in the 2018/2019 season and that fruits maintained or increased TA in response to the increment of K applications (Table 3). Otherwise, results obtained in the Puerto Octay orchard showed that, similar to the observations in the Perquenco orchard, fruits grown under the intensive K regime showed higher firmness at post-harvest compared to fruits from trees cultivated under the conventional K regime; however, these differences were detected only for the 2020/2021 season, whereas, in the 2019/2020 season, no variations were found. Concerning TSS, results indicated that the increase of K applications maintained or increased TSS values in fruits at post-harvest and that TSS levels varied between seasons, being higher in the 2019/2020 season. Also, here we showed that the TSS of fruits at post-harvest did not significantly vary among localities (Table 3).

Regarding the impact of the treatments on the incidence of the most frequent defects of fruits at post-harvest (Table 3), our results revealed that the main defect detected was pitting, followed by pedicel browning and cracking. In addition, our data showed that fruits produced in the Puerto Octay orchard exhibited a higher incidence of defects than fruits from the Perquenco orchard. Moreover, fruits produced in the Perquenco orchard on un-covered trees showed greater cracking and pedicel browning levels than fruits produced in covered trees but also showed a lower incidence of fruit pitting. Furthermore, in the Perquenco orchard, fruits from trees grown under the intensive K regime showed lower pitting incidence than those cultivated under the conventional K regime, a response that was detected in fruits harvested from both un-covered and covered trees and in both study seasons (2018/2019 and 2019/2020). In contrast, no significant variations between K treatments were observed regarding cracking nor pedicel browning in the fruits of the Perquenco orchards. Also, we found that fruits produced in the Perquenco site during the 2019/2020 season presented a higher incidence of cracking than those from the 2018/2019 season. Otherwise, fruits produced in the Perquenco orchard in 2018/2019 showed a greater incidence of fruits with brown pedicels than those produced in 2019/2020. On the other hand, in the Puerto Octay orchard, differences for fruit condition defects were observed between the two seasons, detecting higher pedicel and pitting incidence during the 2020/2021 season compared to the 2019/2020 season.

In agreement with the results observed in the fruits of the Perquenco orchard, additional K applications were capable of reducing the incidence of post-harvest pitting in the fruits of the Puerto Octay orchard. Finally, results indicated that fruits from Puerto Octay trees grown under the intensive K regime showed higher pedicel browning levels than those produced in trees cultivated under the conventional K regime, without significant impacts observed for fruit cracking.

## 3. Discussion

In the last few years, sweet cherry production in Southern Chile has increased due to the negative impacts of climate change on the central zone of this country, such as reductions in water availability, as well as due to the possibility of obtaining better prices for the fruits of late-maturity varieties, which are highly prized in China. Nonetheless, the cultivation of sweet cherry in the southern zone (temperate rainy climate) is frequently affected by several climatic constraints, including the high incidence of rains at bloom and pre-harvest, which forces producers to use plastic covers to mitigate flower falls and, most importantly, fruit cracking. The impact of the use of plastic covers on sweet cherry has been poorly explored; however, some studies have shown that its use can increase the caliber and reduce the cracking [41,42] of fruits but decrease cherry firmness [18,21,22,25] and soluble solids contents [43,44]. On the other hand, according to information provided by the fruit export company Ranco Cherries of Chile, the cherry produced in southern Chile exhibits low acidity levels, which worsens with the use of covers, a factor that is stronger in more southern regions. In this context, studies have shown that potassium (K) fertilization can increase the fruit acidity of sweet cherry and other fruits [28,38], as well as increase fruit firmness [28,38,45]; nonetheless, to date, these findings have not been probed for foliar fertilization practices. Considering these antecedents, the goal of the present study was to evaluate the impact of pre-harvest K foliar applications on the quality and condition of fruits produced in sweet cherry cultivated under a plastic cover in the orchards of southern Chile.

The use of protective plastic covers is recognized as an effective technology to reduce rain-induced fruit cracking in sweet cherries; however, there is a lack of information concerning the effects of this production system on the fruit’s mineral concentration, quality, and post-harvest life. Therefore, we study the impact of two regimes of K foliar fertilization at pre-harvest, a conventional regime (four sprays) and an intensive regime (seven sprays), applied weekly from the phenological stage of straw-yellow fruit color skin phenological stage to 1 week before harvest, in covered orchards of sweet cherry (cultivar Regina) in two localities of southern Chile (Perquenco and Puerto Octay). In addition, the influences of plastic cover on fruit quality and condition were assessed in the Perquenco orchard, due to the fact that, in this locality, the experimental assay considered rows with and without plastic covers.

As expected, our results revealed that the use of plastic covers significantly reduced the incidence of cherry cracking (until ~20%), increasing the proportion of fruits with export-quality characteristics but reducing their firmness (at least 13%) and acidity (between 10% and 20%). Interestingly, we showed that the negative impacts of covers on fruit quality were partially overcome in response to an intensive regime of K foliar fertilization (seven applications during the season) compared to the conventional K regime (four applications during the season.

It is well accepted that fruit firmness is one of the most relevant quality parameters for soft fruits such as sweet cherries [46,47], which normally decrease during maturation due to the progressive degradation of the cell walls of the fruit cells [48]. In this work, in order to assess the impact of light exposure on fruit quality, fruits of the upper and lower area of the canopy were evaluated separately. As a result, we detected firmer fruits in the lower area of the canopy, independent of the treatments, which had already been observed in previous studies in a temperate climate [21,25]; this could be in response to a delay in the maturation of fruits located in the lower area of the canopy, which is highly shaded by the upper area. Reduced sweet cherry fruit firmness by the use of plastic covers detected here, which has also been previously reported [18,21,22], could be explained by the lower levels of calcium (Ca) detected in fruits grown under plastic covers [25].

Calcium is involved in the resistance of cell walls to cracking and excess cell turgor [49], and its flow towards the fruits is decreased due to the restrictions of the transpiration stream [50], provoked by the higher relative humidity commonly detected in orchards protected with plastic covers [21,51]. The higher temperature in covered orchards compared to non-covered ones [17,21,52], provoked by a faster accumulation of hotter days [52], can also explain the low firmness of cherries from covered trees, due to the acceleration of fruit maturation. In the present work, the impact of K applications on fruit firmness was not clear due to that the intensification of the fertilization program (from four to seven applications) revealed contrasting results between both seasons in the Puerto Octay orchards and a positive effect (increased fruit firmness) only during the second season of study in the Perquenco orchard (Table 2). To our knowledge, there are no previous reports about the effect of K applications on the firmness of sweet cherry fruits; however, studies have shown an increase of the firmness in apples [38] and pears [41] provoked by K fertilization, and it has even been reported that post-harvest potassium permanganate applications could improve the apricot post-harvest [45]. However, all these studies were performed in orchards without covers.

In accordance with earlier reports [21,53], herein, we found that plastic cover increased the weight and size of sweet cherry fruit by at least 10%. According to [22], the increased rate division and expansion of fruit cells can be provoked by the higher temperatures under covered conditions, especially in the early stages of fruit maturation. Moreover, our results revealed that the intensification of K applications had a beneficial effect on the weight and size of cherries, such as being previously observed in apricot with post-harvest potassium permanganate applications [45], which can be explained by a high rate of carbohydrate translocation from leaves to fruits induced by K [21], this being one of the main physiological functions of this macronutrient [54,55]. In the present study, the harvest date of both localities presented a difference of almost 2 weeks, being earlier in the Perquenco (38°24′9.39″ S, 72°30′49.38″ W; 250 m.a.s.l) compared to the Puerto Octay orchard (40°52′58″ S; 72°50′15″ W; 150 m.a.s.l.). This harvest lag would be explained by an earlier bloom and faster phenological succession as well as by a faster accumulation of soluble solids during maturation in response to more appropriate climatic conditions for photosynthesis detected for the sweet cherry trees of the Perquenco orchard. According to [56], the use of covers in the cherry orchard delays fruit harvesting by four days beyond the current commercial harvest, achieving higher fruit quality. Soluble solids are one of the most relevant parameters used to define the harvest date in sweet cherries. Besides providing flavor to the fruits, it has an important role in hormonal regulation during the ripening process, promoting the synthesis of anthocyanins and the softening of fruits [57]. Our study showed that the fruits produced in orchard protected with plastic cover exhibited lower soluble solid content than those collected from un-covered orchards. This impact could be explained by the lower incidence of global solar radiation under plastic covers, and the larger size of fruits produced in covered orchards [58], which causes a dilution effect inside the fruit, as was previously reported by [21].

Fruit acidity, defined as the percentage of organic acids inside the pulp [59], is another chemical quality parameter of high importance for fresh fruits, frequently associated with a better taste for the consumer. In addition, fruit acidity contributes to maintaining a good phytosanitary condition of fruits at post-harvest [39]. Malic acid is the main organic acid in sweet cherry fruits [60], and the metabolism of this molecule is strongly influenced by genetic control and climatic conditions [30]. Both the accumulation and subsequent degradation of malic acid from the phenological stage of stone hardening beyond are significantly influenced by ambient temperatures [61]. High temperatures promote the glycolysis and tricarboxylic acid cycles, modifying the enzymatic activities involved in both processes and reducing fruit activity [62]. Despite these antecedents, the results of the present work showed that fruits produced at the Puerto Octay orchard, a locality with lower maximal and minimum temperatures), were less (0.45% to 0.69% of malic acid) than those detected in fruits collected from the Perquenco orchard (0.84% to 1.08% of malic acid). The differences detected in the maturation period between both locations, which was longer in the Puerto Octay orchard compared to the Perquenco orchard, could allow the greater degradation of organic acids. Furthermore, here we found that trees cultivated under plastic cover produced fruits with lower acidity levels than uncovered trees. Besides, results indicated that fruits from trees treated with the intensive K regime have higher acidity levels than those subjected to the conventional K regime, mainly in those protected with plastic cover. To date, there is no consensus about the influence of K nutrition on fruit acidity [30]; moreover, there are no previous reports regarding the impact of K foliar applications on sweet cherry quality and conditions produced under plastic covers. The differences in the maturity index, constructed from the relationship between TSS and TA, between both localities would be directly subordinate to the low levels of TA observed in Puerto Octay with respect to Perquenco. Thus, in general terms, the values found in the first locality are at the lower limit of the ranges reported by other researchers [62]. Etienne [30] discusses the impact of K fertilization on fruit acidity and points out that agronomic observations are contradictory, whereas some authors reported that K supply increased fruit titratable acidity (TA) [63,64]; others concluded that this parameter is decreased in response to K applications [65,66]. Different mechanisms allow K+ to affect the metabolism and storage of organic acids at the cellular level [30]. Firstly, organic anions are synthesized in the vegetative parts to buffer the excess of organic cations absorbed from the soil [67]. Thus, according to Burström [68], fruit supplied with K+ by the sap is necessarily accompanied by an equivalent amount of organic anions, mostly malate, and to a lesser extent, citrate. A modification in fruit TA in response to the supply of K+ implies that K+ affects the synthesis or the vacuolar storage of organic acids within the fruit itself [30]. Furthermore, K+ is known to be involved in the regulation of various enzymes (including the tonoplast proton pumps), either directly [69,70] or by modifying cytosolic pH (Wyn Jones and Pollard, 1983) [69]. However, this is unlikely to play an important role because of the homeostasis of cytosolic K+ [71]. Therefore, Lobit [72] proposes that malate accumulation is controlled by vacuolar pH, only taking the contribution of K+ to the acid–basic reactions in the vacuole into account. This model predicted that, at low vacuolar pH (in the early stages of fruit growth), an increase in K+ content would reduce malate accumulation, while at higher pH (during fruit ripening), it would stimulate it. In grapevines, it is well accepted that K supply can reduce grapes’ acidity. According to Kodur [73], this impact is due to an excessive accumulation of K+, leading to an increase in the electrical neutralization of organic acids, disturbing the pH control and the acid-base balance of the flesh cells.

Wallberg and Sagredo (2014) evaluated the effect of rain protective covers on the vegetative and reproductive development and fruit quality of sweet cherry trees (cultivar Lapins) in Chile [74]. They reported that protective covers filtered approximately 40% of incident photosynthetically active radiation (PAR), stimulating shoot growth and reducing fruit coloration. The protective covering installed at flower bud-burst increased fruit size, weight, and soluble solids content but reduced fruit firmness. Similarly, Blanco et al. [25] studied the feasibility of using high tunnels on ‘Santina’ sweet cherries under the Mediterranean climate of the Central Valley of Chile to obtain earlier harvests of high-quality fruits with long storage life. They found that high temperatures and relative humidity inside the high tunnels (during bloom and fruit set) decreased fruit yield and that trees inside the high tunnels were harvested 11 days earlier. Also, the same authors shown that fruits from covered trees was significantly larger and softer than outside ones, nonetheless, high-tunnel did not affect TSS not TA of cherries The same authors found that fruits harvested from the lower area of the canopy exhibited lower values of and TSS but higher TA, firmness, weight, and size.

Regard the possible mechanism explaining the effect of K fertilization on fruit quality, several hypothesis are proposed. It is well-known that K plays a key role in photosynthesis and the regulation of the opening and closing of the stomata, favoring high-energy status, which helps in suitable water uptake and nutrient translocation in plants [75]. In addition, ref. [76] stated that K is one of the main nutrients essential for plant growth and development and that it activates enzymes important to energy utilization, starch synthesis, nitrogen (N) metabolism, and respiration. Also, it has been reported that the application of adequate K doses could increase fruit weight by improving photosynthates’ translocation to fruit and water-use efficiency [77]. The significant role of K in carbohydrate formation and the transformation and movement of photosynthates from sites of production to storage organs has also been reported [78,79,80,81].

Moreover, it has been demonstrated that K fertilization in fruit crops, playing a structural and osmoregulatory role, modulates the expression of genes capable of promoting the process of growth and ripening and the formation of fruits with a higher quality (concerning fruits with a deficit of this nutrient). The mechanisms explaining the positive impacts of K on fruit quality and condition are complex, implying the regulation of at least five plant hormones and several transporters [82]. On the other hand, Amjad et al. [83] showed that K supply significantly increases fruit diameter in tomatoes, suggesting that this impact could be ascribed to the activation of enzymes and its involvement in adenosine triphosphate (ATP) production, which is important in regulating the photosynthesis rate that allows plants to have more food to be stored in the fruits [76] (Havlin et al., 2005). In this context, ATP is also used as the energy source for many plant activities [84], including cell divisions that determine the final fruit size [85].

The main obstacle in cherry production is fruit cracking induced by rain. This condition defect causes severe losses in many production areas of sweet cherry [19,86,87]. Cracking is frequently the result of a wet fruit surface prior to harvest; therefore, water uptake through the skin and pedicel results in fruits provoking an increase in turgor, inducing cracking [88,89]. In this regard, the cultivation of sweet cherry under plastic covers is considered one of the best methods to avoid the rain-induced cracking of fruits [90,91]. Our results revealed that fruit cracking was the main defect detected in the fruits of our assays, independent of the location and treatments applied. In agreement with previous studies [18,25,43,44], we found that cracking incidence was lower in fruits from trees protected with the plastic cover than un-covered ones. Despite the sweet cherry variety used here, which is described as cracking-resistant [43,44], the levels of cracking detected reached values ranging from 2% to 10% in the covered plot and from 3% to 20% in the un-covered plot, which was decreased in response to the intensive K regime compared to the conventional K regime. These results are not far from those reported for the sensitive varieties Brooks and Royal Dawn, in which cracking losses can be even higher than 40% when trees are cultivated without covering [41]. The higher levels of cracking detected here in fruits from covered trees compared to un-covered ones can be explained by the higher exposure of cherries to the precipitation and condensation of environmental water prior to harvest [41,42], as well as to their higher fruit firmness, which increases susceptibility to cracking in cherries [92]. Blanco et al. [25] reported no differences between covered and uncovered cherries in either cracking susceptibility or induced pitting on the fruits of sweet cherry cv. Santina cultivated in the central valley of Chile. On the other hand, cracking incidence during post-harvest storage can increase when fruits are exposed to water contact on the cuticle during fruit ripening, which provokes the generation of microscopic fissures in fruit skin [49]. Our results revealed no significant differences in the incidence of cracking between the different regimes of K fertilization, neither in pre-harvest nor in post-harvest. According to [92], nitrogen and K would decrease the susceptibility to cracking. In addition, we found that cracking incidence tends to be higher in the Puerto Octay location than the Perquenco orchard.

The browning and dehydration of pedicels are important symptoms of loss of quality and are considered indicators of post-harvest freshness for sweet cherry fruits [40]. Herein, we found a reduction of browning pedicel incidence in fruits from trees cultivated under plastic cover compared to the uncovered ones. It has been reported that the presence of water over the fruits can provoke the crack of the pedicels’ epidermic cells, then triggering the drying process [93] and the enzymatic degradation of chlorophylls in this structure [94]. Herein, we found that brown pedicels were more common on un-covered plots, an effect that was decreased by the impact of the intensive K regime. Another frequent condition defect of cherry fruits is pitting, defined as surface depressions in the fruit surface caused by the collapse of cells under the skin provoked by either harvest, sorting, or packaging processes and even throughout post-harvest storage [95]. In the present study, fruits collected from un-covered orchard exhibited lower pitting incidence than covered ones, related to the lower firmness of fruits from trees cultivated under plastic covers, reducing the incidence of pitting in cherries [21,25]. Interestingly, the intensive regime of K application decreased the incidence of this fruit condition symptom at post-harvest, which agrees with [96], reporting lower susceptibility of pitting cherry fruits in response to high doses of K compared to low doses.

## 4. Materials and Methods

### 4.1. Plant Material, Treatments and Experimental Design

Field studies were carried out in two commercial orchards of sweet cherry (*Prunus avium* L.) of cultivar Regina grafted on Gisela 6 rootstock, located at two different regions of southern Chile. The geographical locations of field assays and sweet cherry cultivation area can be seen in Figure 4.

The first field assay was established in an orchard located at La Araucanía Region (Perquenco locality; 38°24′9.39″ S, 72°30′49.38″ W; 250 m.a.s.l.). This experiment was performed during two consecutive seasons (2018/2019 and 2019/2020). In this commercial farm, trees were planted in 2012 in a 4.5 m × 1.6 m planting design, reaching a planting density of 1389 trees ha^−1^, using an east-west row orientation. The climate of this location, the minimum, maximum, and average temperatures were all recorded from bud to harvest period (November to February) at the nearest weather station (San Sebastián station, INIA) [97]; they were 6.8 °C, 22 °C, and 14.5 °C, respectively, whereas the average of relative humidity (HR) recorded was 69%. Thus, temperature and HR conditions were very similar during both seasons; however, accumulated precipitation registered in the same period was higher (~40%) during the 2018/2019 season (148 mm) compared to the 2019/2020 season (85 mm).

The second field assay was established in an orchard located at Los Lagos Region (Puerto Octay locality; 40°52′58″ S; 72°50′15″ W; 150 m.a.s.l.). The experiment was performed during the 2019/2020 and 2020/2021 harvest seasons to compare to the results obtained in Perquenco, since this locality is further south. In this farm, trees were planted in 2014 in a 4.5 m × 2.0 m planting design, reaching a planting density of 1111 trees ha^−1^, using a north-south row orientation. The climate of this farm showed 9.5 °C as the minimum, 18.5 °C as a maximum, and 14 °C as average temperatures in the 2020/2021 season during the bud to harvest period (November–February) at the nearest weather station (Puerto Octay station, INIA) [97], whereas the average of relative humidity recorded was 74%. Temperature and HR conditions were similar during both seasons; however, as occurred at Perquenco, accumulated precipitation in the same period was higher (~40%) during the 2019/2020 season (281 mm) than during the 2020/2021 season (170 mm).

In both orchards, the plastic covers used consisted of a movable high-density polyethylene (HDPE) plastic in gable form and have the following properties: 4 meters of width, an average density of 160 g m^−2^, 88–90% of total light transmission, and 60–65% of diffuse light transmission. In the Perquenco orchard, covers were deployed (close) from the bloom start to the fruit set phenological stages and then 2 weeks before harvest, whereas in the Puerto Octay orchard, covers were deployed throughout the growth period. In this last location, covers are essential to protect trees against high levels and intensity of rainfall. For the same reason, in the Puerto Octay assay, it was impossible to assess the impact of K foliar applications on un-covered trees because all trees were protected (covered) in the experiment.

In both assays, foliar applications of potassium (K) were carried out by using a bag bomb (Cifarelli^®^, Model L3EDA, Voghera, Italy). Each K application was applied by spraying 4 L ha^−1^ of the chelated fertilizer Nutripotasio Plus (ADAMA^®^; 38% of K_2_O; diluted in water) to reach a final concentration of 300 mL hL^−1^. For the treatment identified as conventional K regime, foliar applications were applied starting in the straw-yellow fruit skin color phenological stage, weekly, for 4 weeks, reaching a total of 4 sprays, whereas, for the treatment identified as intensive K regime, 3 additional sprays were applied, weekly, from the last application of the conventional K regime to 1 week before harvest, reaching 7 sprays. Then, in this study, a total of 12 treatments were applied, considering the interaction of location × season × covering × K applications. The details of treatments applied can be seen in Table 4.

In the Perquenco orchard, the experimental design used was a split-plot; one plot consisted of covered trees, and another plot consisted of un-covered trees. Each plot was divided into five blocks per K treatment, each block (replicate) corresponding to a row of 4 consecutive cherry trees. Otherwise, in the Puerto Octay orchard, a completely randomized block design was used, also considering five blocks (a row of 4 consecutive cherry trees) per each K treatment. In both experiments, quality and condition parameters were analyzed only in fruits collected from the 2 central trees of each block, whereas the 2 remaining trees were considered border plants. A scheme of the experimental design used in this work can be seen in Figure 5.

### 4.2. Evaluation of Quality and Condition of Fruits at Harvest

A total of 200 fruits per tree were collected from the 2 central trees of each row, considering both tree exposures. In order to evaluate the impact of the light exposure of trees on the different parameters studied herein, 100 fruits were separately harvested from the upper area (height > 1.2 m) and the lower area (height < 1.2 m) of the canopy. Once harvested, fruits were put in 1 kg plastic bags and transported (4 °C) to the Fruit Crops Physiology and Quality Laboratory of La Frontera University. Then, the fruits were stored at 4 °C prior to measuring quality and conditions parameters within 48 h after harvest. Thus, 5 independent samples were analyzed per each treatment. The physical quality parameters of weight, equatorial diameter (size), and firmness of 35 fresh fruits per sample were measured, using an analytical balance and a texture-meter (FirmPro, Happyvolt, Santiago, CL). This equipment can automatically measure fruit caliber and firmness at the same time, which has been validated for fruit firmness determination in sweet cherry cultivar Lapins [98]. Also, herein, we measured total soluble solids (TSS), titratable acidity (TA), and maturity index (MI) as chemical quality parameters. Briefly, TSS was analyzed in the fruit juice using a thermo-compensated digital refractometer (ATAGO, Mod. PAL-BX I ACID F5, Saitama, Japan) and expressed as ºBrix. In addition, TA was determined by the volumetric titration method with sodium hydroxide (0.1 N), using an automatic titrator (HANNA Mod. HI-84532, Woonsocket, US), and expressed as a percentage (%) of malic acid. Finally, the maturity index was calculated as the ratio between TSS and TA. A trained panel qualitatively and visually estimated the most frequent fruit condition defects, including cracking, browning pedicels, and pitting, which were analyzed previously to determine physical parameters.

### 4.3. Analysis of Quality and Condition of Fruits at Post-Harvest

In order to evaluate the quality and condition of fruits at post-harvest, 3 harvest plastic boxes (~10 kg capacity) of fruits from the 5 replicates per treatment were collected and sent to the Export Company Ranco Cherries for storage under real export conditions. In all cases, fruits were harvested from the 2 central trees of each block, including all canopy areas. Briefly, once they arrived at the Export Company, fruits were hydro-cooled to reduce pulp temperature to around 3–4 °C and kept in a cold-chamber for 24 h. Then, cherries were manually packed, discarding only cherries with evident gross defects, and immersed in a fungicide (Fludioxonil) solution. The packaging used for cherry storage was 5-kg modified atmosphere plastic boxes (5–6% CO_2_; 16–17% O_2_). Then, the sealed boxes were exposed to a Californian-type forced-air tunnel to reduce fruit temperature to 0 °C and stored for 30 days at 0 °C and relative humidity of 90% in a cold chamber. Prior to analysis, fruits were acclimatized for 16 h at 20 °C. For firmness and condition determination, 34 and 50 fruits per box were individually measured, whereas, for TSS and TA, 3 different samples per treatment were measured, each sample consisting of a juice obtained by macerating 10 fruits from each box.

### 4.4. Statistical Analysis

Data were analyzed by analysis of variance (ANOVA) using Infostat^®^ Software Version 2017, following normality and homoscedasticity determined by Shapiro Wilks and Levene’s tests, respectively. Significantly differences between the means were evaluated by using the LSD Fisher multiple comparison tests (*p* ≤ 0.05). Additionally, a principal component analysis (PCA) was done to identify the variables that explained the relationship between fruit quality parameters and treatments applied. Also, the proportion of variance explained (estimated as the sum of squares of the treatment over the total sum of squares) for the factors involved in this study was analyzed

## 5. Conclusions

The findings in our study contribute to clarifying the impact of K fertilization practices applied by cherry farmers in southern Chile to improve the quality and condition of sweet cherry fruits. The main results of this work allow us to conclude that, despite plastic covers being very useful to reduce fruit cracking in Regina sweet cherry fruits in both southern Chile orchards, increasing the level of export quality fruits, tree covering can also provoke important reductions of cherry firmness and acidity both at harvest and post-harvest. The intensive K foliar regime, applied from the straw-yellow fruit skin color phenological stage, can partially mitigate these negative impacts; however, the impacts can be strongly dependent on season and locality. Furthermore, herein, we found that condition defects were lower in fruits from covered trees and that trees grown under intensive K regime showed lower incidence of cracking at harvest and pitting at post-harvest compared to trees treated with the conventional K regime. Further studies are needed to elucidate the physiological mechanisms that explain K effects, as well as the K absorption and distribution in plant tissues in response to the K treatment, in order to contribute to the scientific understanding of the important findings with regard to the benefits of intensive K foliar application on sweet cherry produced under covered orchards.

## Figures and Tables

**Figure 1 plants-10-02778-f001:**
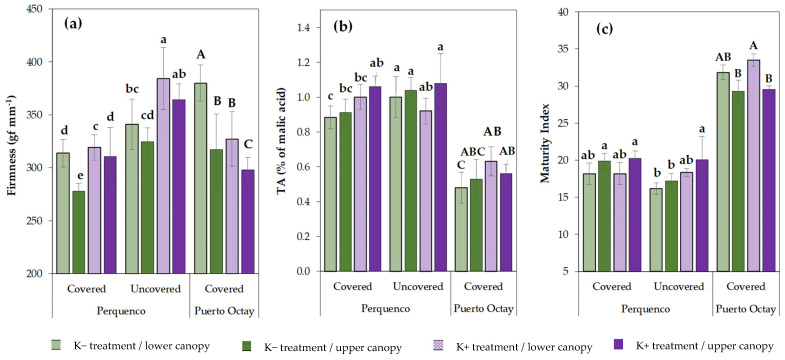
Firmness (**a**), titratable acidity (**b**), and maturity index (**c**) at harvest of sweet cherry fruits (cultivar Regina) harvested from trees cultivated in two commercial orchards of southern Chile (Perquenco and Puerto Octay, during the 2019/2020 season. In each location, trees were subjected to two foliar potassium (K) treatments: conventional K regime (4 sprays during the season: K− treatment) or intensive K regime (7 sprays during the season; K+ treatment). In Perquenco orchard, covered and un-covered trees were compared. Fruits harvested from the lower or the upper area of the canopy were evaluated separately. Different lowercase and uppercase letters above the error bars (standard deviation) indicate significant differences between the treatments in the Perquenco and Puerto Octay orchards, respectively, according to LSD Fisher multiple range test (*p* ≤ 0.05).

**Figure 2 plants-10-02778-f002:**
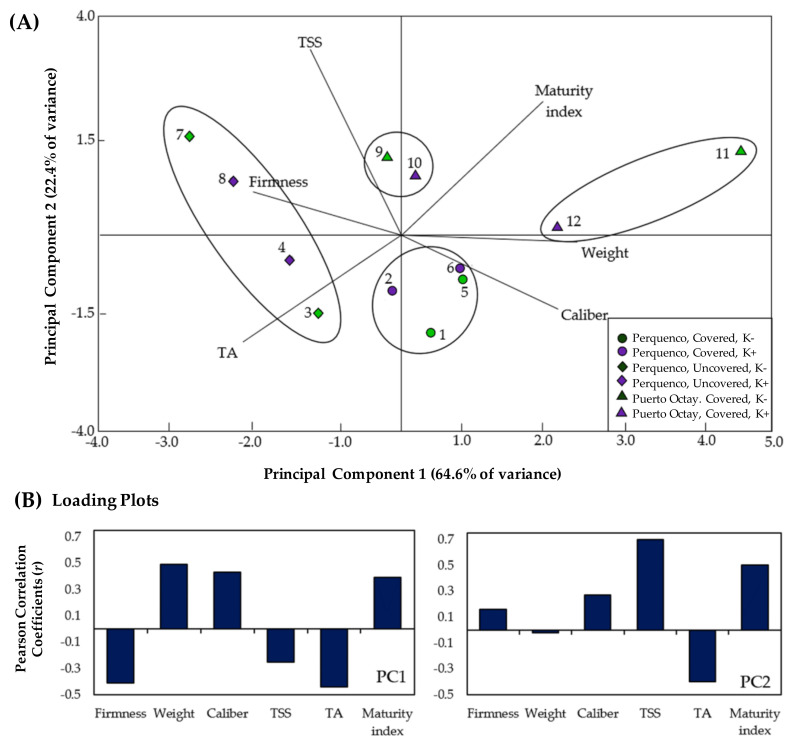
Results of principal component analysis (PCA): (**A**) dispersion plot of PC1 versus PC2 and (**B**) loading plots of PC1 and PC2, representing the main quality parameters of sweet cherry fruits based on different localities (Perquenco and Puerto Octay), seasons, use of plastic covers, and potassium (K) foliar fertilization regimes. Trees of sweet cherry (cultivar Regina) were subjected to two foliar potassium (K) treatments: a conventional K regime (4 sprays during the season: K− treatment) or an intensive K regime (7 sprays during the season; K+ treatment). In each location, the assays were conducted in two consecutive seasons. In the Perquenco orchard, covered and un-covered trees were compared. The points within the scatter plot are: Perquenco orchard, 2018/2019 season, covered, treated with K− regime (1), Perquenco orchard, 2018/2019 season, covered, treated with K+ regime (2), Perquenco orchard, 2018/2019 season, un-covered, treated with K− regime (3), Perquenco orchard, 2018/2019 season, un-covered, treated with K+ regime (4), Perquenco orchard, 2019/2020 season, covered, treated with K− regime (5), Perquenco orchard, 2019/2020, covered, treated with K+ regime (6), Perquenco orchard, 2019/2020 season, un-covered, treated with K− regime (7), Perquenco orchard, 2019/2020 season, un-covered, treated with K+ regime (8), Puerto Octay orchard, 2019/2020 season covered, treated with K− regime (9), Puerto Octay orchard, 2019/2020 season, covered, treated with K+ regime (10), Puerto Octay orchard, 2020/2021 season, covered, treated with K− regime (11), and Puerto Octay orchard, 2020/2021 season, covered, treated with K+ regime (12). Abbreviations: TSS (total soluble solids), AT (titratable acidity).

**Figure 3 plants-10-02778-f003:**
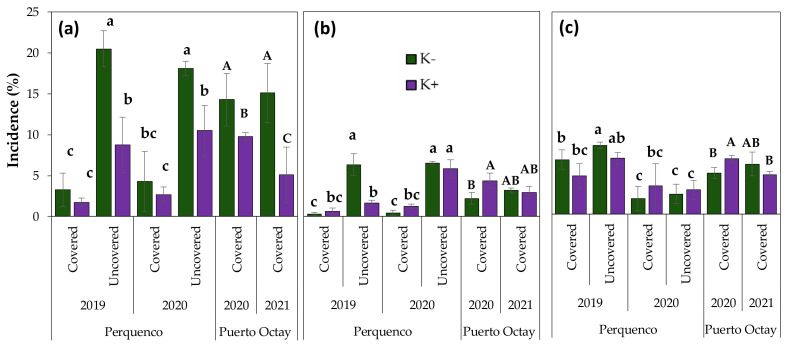
Incidence (%) of cracking (**a**), pedicel browning (**b**), and pitting (**c**) at harvest in fruits of sweet cherry (cv. Regina) cultivated at two orchards of southern Chile (Perquenco and Puerto Octay), subjected to two foliar potassium (K) treatments: conventional K regime (4 sprays during the season: K− treatment, white bars) or intensive K regime (7 sprays during the season; K+ treatment; black bars). In each location, the assays were conducted in two consecutive seasons. In the Perquenco orchard, covered and un-covered trees were compared. Different lowercase and uppercase letters above the error bars (standard deviation) indicate significant differences between the treatments in the Perquenco and Puerto Octay orchards, respectively, according to LSD Fisher multiple range test (*p* ≤ 0.05).

**Figure 4 plants-10-02778-f004:**
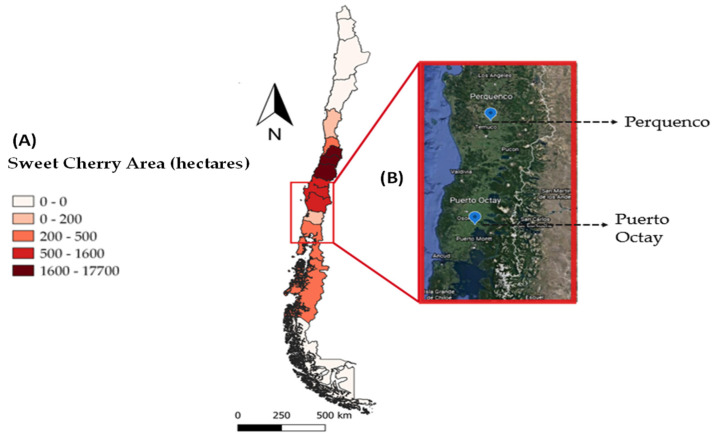
Distribution of the sweet cherry planting area in Chile (Source: ODEPA, 2021) (**A**) and location of the study orchards at southern Chile (**B**).

**Figure 5 plants-10-02778-f005:**
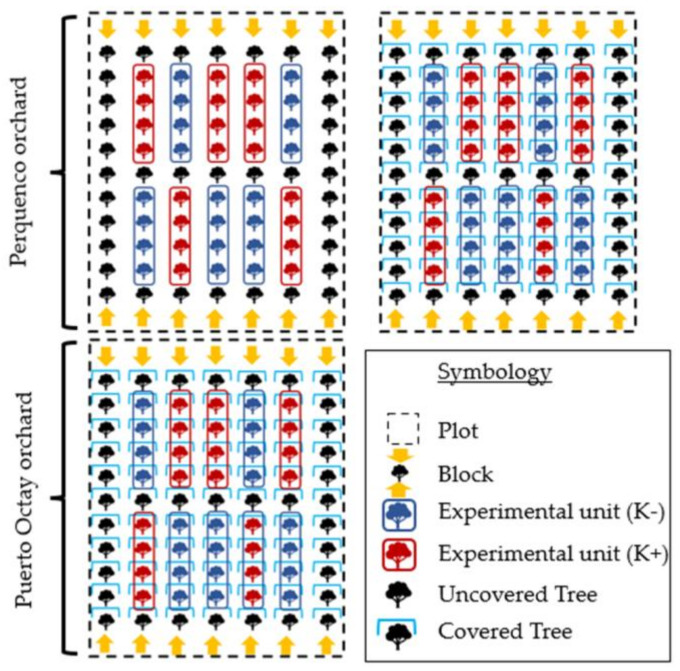
Experimental design of the field study performed in two sweet cherry commercial orchards located at different localities of southern Chile: Perquenco and Puerto Octay. Two foliar potassium (K) treatments were applied: conventional K regime (4 sprays during the season: K− treatment) or intensive K regime (7 sprays during the season: K+ treatment). In each location, the assays were conducted in two consecutive seasons. In the Perquenco orchard, covered and un-covered trees were compared.

**Table 1 plants-10-02778-t001:** Fruit quality parameters at harvest in fruits of sweet cherry (cultivar Regina) cultivated in two commercial orchards located in southern Chile (Perquenco and Puerto Octay), subjected to two foliar potassium (K) treatments: conventional K regime (4 sprays during the season: K- treatment) or intensive K regime (7 sprays during the season; K+ treatment). In each location, the assays were conducted in two consecutive seasons. In the Perquenco orchard, covered and un-covered trees were compared. Abbreviations: TSS (total soluble solids), TA (titratable acidity).

Treatment							Significance		
Farm	Season	Fruit Parameter	Covered		Uncovered		Potassium	Cover	Cover × Potassium
			K−	K+	K−	K+			
Perquenco	2019	Firmness (g mm^−1^)	296 ± 22 Cb	315 ± 21 BCa	333 ± 22 BCb	375 ± 25 Aa	***	***	ns
	2020		298 ± 32 Bb	303 ± 31 Ba	356 ± 30 Aa	344 ± 28 Ab	ns	***	ns
Puerto Octay	2020		349 ± 45 Aa	313 ± 37 Ba			*		
	2021		269 ± 21 Bb	324 ± 17 Aa			***		
**Season-locality significance**			******	**ns**	******	*****			
Perquenco	2019	Weight (g)	10.2 ± 0.9 Ab	10.3 ± 0.9 Ab	9.9 ± 1.2 Aa	9.7 ± 1.0 Aa	ns	ns	ns
	2020		10.7 ± 0.6 Ab	11.0 ± 0.8 Ab	8.7 ± 1.8 Bb	9.2 ± 0.3 Ba	ns	***	ns
Puerto Octay	2020		10.2 ± 0.6 Ab	10.3 ± 1.3 Ab			ns		
	2021		12.1 ± 0.8 Aa	11.8 ± 0.9 Aa			ns		
**Season-locality significance**			******	******	**ns**	**ns**			
Perquenco	2019	Caliber (mm)	27.1 ± 0.5 ABb	27.5 ± 0.9 Aab	26.7 ± 0.9 Ba	27.3 ± 0.5 Aba	*	ns	ns
	2020		27.5 ± 0.7 Aab	27.8 ± 0.8 Aa	25.7 ± 0.6 Bb	26.0 ± 0.5 Bb	ns	***	ns
Puerto Octay	2020		27.1 ± 0.7 Ab	26.6 ± 1.3 Ab			ns		
	2021		28.1 ± 0.9 Aa	28.0 ± 1.0 Aa			ns		
**Season-locality significance**			*****	*****	******	*******			
Perquenco	2019	TSS (Brix)	17.0 ± 1.8 Cc	19.2 ± 1.0 ABb	18.0 ± 1.9 BCb	19.5 ± 1.1 Ab	*	ns	ns
	2020		18.9 ± 2.0 Bb	19.9 ± 2.2 Bab	22.9 ± 1.9 Aa	21.6 ± 1.2 Aa	ns	***	ns
Puerto Octay	2020		21.2 ± 1.6 Aa	20.6 ± 1.8 Aa			ns		
	2021		19.0 ± 0.9 Ab	19.1 ± 1.2 Ab			ns		
**Season-locality significance**			*******	**ns**	*******	*******			
Perquenco	2019	TA (% malic acid)	0.90 ± 0.1 Ba	1.00 ± 0.1 Aa	1.08 ± 0.1 Aa	1.03 ± 0.1 Aa	***	ns	**
	2020		0.84 ± 0.1 Ca	0.91 ± 0.1 BCb	1.02 ± 0.2 Aa	1.00 ± 0.1 ABa	*	***	ns
Puerto Octay	2020		0.69 ± 0.1 Ab	0.69 ± 0.1 Ac			ns		
	2021		0.45 ± 0.1 Bc	0.60 ± 0.1 Ad			***		
**Season-locality significance**			*******	*******	**ns**	**ns**			
Perquenco	2019	Maturity index	19.0 ± 2.6 Ac	19.2 ± 1.6 Ac	16.7 ± 1.1Bb	19.2 ± 2.6Ab	*	ns	ns
	2020		22.5 ± 1.5 Ac	21.9 ± 1.8 Ab	22.8 ± 3.6Aa	21.7 ± 1.2Aa	ns	ns	ns
Puerto Octay	2020		30.6 ± 2.0 Ab	29.9 ± 0.7 Aa			ns		
	2021		42.1 ± 2.1 Aa	32.1 ± 3.7 Ba			**		
**Season-locality significance**			*******	*******	*******	******			

Statistically significant differences between treatments for each season and orchard are represented by different uppercase letters (horizontally), whereas differences between season and orchard for each K treatment are represented by vertical lowercase letters, based on the LSD Fisher multiple range test (*p* ≤ 0.05). Asterisks indicate a significant interaction between the factors (* *p* ≤ 0.05, ** *p* ≤ 0.01, *** *p* ≤ 0.005).

**Table 2 plants-10-02778-t002:** Percentage (%) of variance explained by the effect of the independent variables for the most relevant fruit quality parameters of sweet cherry (cultivar Regina) cultivated in two commercial orchards located in southern Chile (Perquenco and Puerto Octay), subjected to two foliar potassium (K) treatments: conventional K regime (4 sprays during the season: K- treatment) or intensive K regime (7 sprays during the season; K+ treatment). In each location, the assays were conducted in two consecutive seasons. In the Perquenco orchard, covered and un-covered trees were compared. Abbreviations: TSS (total soluble solids); TA (titratable acidity); MI (maturity index).

Perquenco					Puerto Octay		
Parameter	Season	Covers	K Treatment	Canopy Area	Season	K Treatment	Canopy Area
Firmness	0.3	45.6	3.5	4.6	17.5	0.3	13.9
Weight	0.1	30.6	0.1	0.4	37.8	0.1	0.7
Caliber	3.8	28.3	4.2	0.1	24.7	1.7	0.1
TSS	24.5	13.3	3.0	11.2	30.0	0.8	17.2
TA	5.8	24.2	0.8	1.7	51.9	12.3	1.2
Maturity index	43.5	0.9	0.2	3.8	27.8	18.2	2.4

**Table 3 plants-10-02778-t003:** Quality and condition parameters at post-harvest in fruits of sweet cherry (cultivar Regina) cultivated at two commercial orchards of southern Chile (Perquenco and Puerto Octay). In each orchard, trees were subjected to two foliar potassium (K) treatments: conventional K regime (four sprays during the season; K− treatment) or intensive K regime (seven sprays during the season; K+ treatment). In each location, the assays were conducted in two consecutive seasons. In Perquenco orchard, covered and un-covered trees were compared.

Orchard	Season	Fruit Parameter	Treatments			
			Covered		Un-Covered	
			K−	K+	K−	K+
Perquenco	2019	Firmness (g/mm)	327 ± 18 Db	369 ± 18 Cb	418 ± 22 Bb	471 ± 18 Aa
	2020		468 ± 14 Ba	458 ± 34 Ba	493 ± 42 Aa	479 ± 20 Aa
Puerto Octay	2020		476 ± 49 Aa	441 ± 24 Aa		
	2021		281 ± 20 Bc	325 ± 23 Ac		
Perquenco	2019	TSS (Brix)	16.3 ± 0.9 Bb	18.4 ± 1.2 Aab	16.6 ± 0.7 Bb	18.1 ± 0.5 Ab
	2020		19.5 ± 0.9 Ba	19.6 ± 1.2 ABa	20.2 ± 2.0 ABa	21.1 ± 1.6 Aa
Puerto Octay	2020		19.9 ± 1.4 Aa	19.1 ± 1.3 Aa		
	2021		15.8 ± 1.1 Bb	17.3 ± 1.4 Ab		
Perquenco	2019	TA (% malic acid)	0.31 ± 0.02 Bc	0.36 ± 0.04 ABb	0.33 ± 0.02 Bb	0.41 ± 0.03 Ab
	2020		0.58 ± 0.09 Ba	0.53 ± 0.05 Ba	0.61 ± 0.06 Aa	0.59 ± 0.02 Aa
Puerto Octay	2020		0.37 ± 0.01 Ab	0.38 ± 0.04 Ab		
	2021		0.39 ± 0.17 Aab	0.37 ± 0.01 Ab		
Perquenco	2019	Cracking (%)	0.4 ± 0.2 Bb	0.8 ± 0.2 Bc	10.7 ± 2.6 Aa	9.2 ± 3.3 Aa
	2020		1.2 ± 0.6 Bb	1.9 ± 1.0 Bab	14.7 ± 5.2 Aa	12.4 ± 3.7 Aa
Puerto Octay	2020		2.9 ± 0.7 Aa	2.8 ± 1.3 Aa		
	2021		2.1 ± 1.2 Aab	1.2 ± 0.2 Ab		
Perquenco	2019	Pedicel Browning (%)	5.0 ± 1.0 Cb	5.7 ± 1.5 Cc	11.7 ± 1.5 Ba	17.3 ± 3.8 Aa
	2020		4.7 ± 1.2 Bb	5.3 ± 2.4 Bc	8.1 ± 1.9 Ab	9.5 ± 3.5 Ab
Puerto Octay	2020		9.3 ± 3.2 Ba	14.4 ± 4.9 Ab		
	2021		14.5 ± 3.8 Ba	24.0 ± 5.9 Aa		
Perquenco	2019	Pitting (%)	16.3 ± 1.0 Aa	4.9 ± 2.5 Bb	7.4 ± 2.7 Aba	1.6 ± 0.2 Bb
	2020		14.1 ± 5.2 Aab	7.3 ± 3.8 Bab	5.4 ± 2.8 Bca	3.1 ± 1.1 Ca
Puerto Octay	2020		12.3 ± 5.4 Ab	8.7 ± 3.6 Ba		
	2021		16.3 ± 1.0 Aa	5.0 ± 2.5 Bb		

For each parameter, average values ± standard deviation are presented. Statistically significant differences between treatments, for each season and orchard, are represented by different uppercase letters (horizontally). Differences between season and orchard, for each K treatment, are represented by vertical lowercase letters based on the LSD Fisher multiple range test (*p* ≤ 0.05).

**Table 4 plants-10-02778-t004:** Treatments evaluated in trees of sweet cherry (cultivar Regina) cultivated in two commercial orchards of southern Chile (Perquenco and Puerto Octay), subjected to two foliar potassium (K) treatments: conventional K regime (4 sprays during the season: K- treatment) or intensive K regime (7 sprays during the season; K+ treatment). In each location, the assays were conducted in two consecutive seasons. In the Perquenco orchard, covered and un-covered trees were compared.

Locality	Season	Use of Covers	Covers Management	N° of K Sprays	K Treatments
Perquenco	2019	Yes	Closed between flowering and fruit set	4	K−
			+2 weeks prior to harvest	7	K+
		no	Without covers	4	K−
				7	K+
	2020	Yes	Closed between flowering and fruit set	4	K−
			+2 weeks prior to harvest	7	K+
		no	Without covers	4	K−
				7	K+
Puerto Octay	2020	Yes	Closed throughout the productive season	4	K−
				7	K+
	2021	Yes		4	K−
				7	K+

## Data Availability

The data presented in this study are available in the article.

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
