# Peer review of "Impact of Potassium Pre-Harvest Applications on Fruit Quality and Condition of Sweet Cherry (Prunus avium L.) Cultivated under Plastic Covers in Southern Chile Orchards"

_plants, 2021, doi:10.3390/plants10122778_

Round 1
Reviewer 1 Report
I have no suggestions.
Author Response
Thanks for your revisions
Reviewer 2 Report
The manuscript presents original results that may be useful for cultivating sweet cherries.
The results are clearly presented and well discussed. However, there are some minor comments:
Keywords should be rewritten. Most of them are also in the title.
1. Introduction
The paragraph on the effect of potassium applications on the properties of different fruit species should be developed and more detailed. (lines 98-100)
4. Materials and Methods
line 642: Why was the final concentration of 300 mL h L-1 chosen to apply?
5. Conclusions
The conclusion section is short. Conclusions are too general. There are no recommendations for future studies.
Author Response
Thanks for your comments and suggestions. All these were considered in the new version of our manuscript.

Reviewer 3 Report
This is a well-conducted research and the findings are of importance to cherry growers and of interest to researchers in the field. Although the results have been thoroughly analysed statistically, the study does not contribute to our scientific understanding of the important findings with regard to the benefits of intensive potassium foliar application. Mention is made of sampling fruit for assay of antioxidant parameters, but no data are presented and even the effect on the fruit potassium content was not studied. There is little attempt in the discussion to speculate or explain the results regarding the mode of action of potassium in its effect on fruit quality.
Specific points that raise questions or need clarification:
Line 31: Not absolutely correct – intensive K application didn’t reduce postharvest cracking and pedicel browning was inconsistently affected, both pre- and post-harvest. However, its significant effect in reducing postharvest pitting is worthy of mention.
Lines 176-178: This statement is not supported by data in Table 1.
Line 181: Not really ‘similar’, as the two years of trial in each orchard differed.
Line 225: What is the meaning of 'un significant'? In 2020, the difference is NS and in 2021 p<0.01.
Lines 229-230: This statement is incorrect according to Figure 1. Moreover, despite the lower acidity in fruit from the lower canopy of covered trees, no significant differences were found in the maturity indices. Can you explain this without providing TSS data’ which are mentioned as significant in Table 2?
Line 232: No data are shown for the 20/21 season.
Figure 2: Suggestion – different symbols for K treatments, and possibly, also for seasons, would help the reader.
Line 295 and Table 2: If ‘canopy part’ and ‘tree harvest area’ are identical, use only one definition. Suggestion – canopy area.
Table 3: Define postharvest conditions in table title, such as storage temperature, humidity, atmosphere and duration, unless you decide to add these particulars to the Materials and Methods section. Incorrect value for TSS in K- treatment in Perquenco orchard in 2019. Check the ABC markings for TA Perquenco orchard in 2019.
Line 426: Citation 40 is incorrect.
Line 496: Do you mean ‘under’ plastic cover instead of ‘over’?
Line 502: Citation 55 is incorrect and is the same as citation 40.
Line 576: Incorrect with regard to pre-harvest cracking, which was significantly reduced due to intensive K application.
Line 594: Incorrect citation.
Lines 645-6: Specify when the 3 additional sprays were applied.
Line 677: Why mention storage of samples at -80C for antioxidant assays that are not shown?
Lines 695-707: Were 10 kg fruit for storage harvested from 3 or 5 orchard replicates? If the former, it is not clear how six 5 kg modified atmosphere boxes were compiled from 5 orchard replicates. If the latter, were ten 5 kg boxes stored per treatment? What was the atmosphere composition within the packages? How long was the fruit held in cold storage? Were the sealed boxes opened at removal from storage or after 16 hours as 20C?
Line 865: Citation 63 is incomplete.
The English style is sometimes difficult to follow, mainly due to grammatical mistakes and missing words. Sometimes you use AT instead of TA. If you resubmit a corrected MS, I suggest you have someone proofread it before submission.
Author Response
Thanks very much for your interesinting suggestions and comments. All these were considered in the new version of our manuscript. Best Regards. Please see in attached file the datails about the corrections applied.
